# Frequency of *Chlamydia trachomatis* and *Neisseria gonorrhoeae* in Patients with Imminent Preterm Delivery on the Island of Curaçao

**DOI:** 10.3390/pathogens11060670

**Published:** 2022-06-09

**Authors:** Aglaia Hage, Naomi C. A. Juliana, Leonie Steenhof, Ralph R. Voigt, Servaas A. Morré, Elena Ambrosino, Nurah M. Hammoud

**Affiliations:** 1Department of Obstetrics and Gynecology, Curaçao Medical Center, Willemstad, Curacao; hage.aglaia@gmail.com (A.H.); naomijuliana@hotmail.com (N.C.A.J.); lsteenhof94@gmail.com (L.S.); voigtralph@gmail.com (R.R.V.); 2Institute for Public Health Genomics (IPHG), Department of Genetics and Cell Biology, Research School GROW for Oncology and Reproduction, Faculty of Health, Medicine & Life Sciences, University of Maastricht, 6229 ER Maastricht, The Netherlands; samorretravel@yahoo.co.uk (S.A.M.); e.ambrosino@maastrichtuniversity.nl (E.A.); 3Department of Molecular and Cellular Engineering, Jacob Institute of Biotechnology and Bioengineering, Sam Higginbottom University of Agriculture, Technology and Sciences, Allahabad 211007, UP, India; 4Dutch *Chlamydia trachomatis* Reference Laboratory on behalf of the Epidemiology and Surveillance Unit, Centre for Infectious Disease Control, National Institute for Public Health and the Environment, 3721 MA De Bilt, The Netherlands

**Keywords:** *Chlamydia trachomatis*, *Neisseria gonorrhoeae*, sexual and reproductive health, pregnancy, imminent preterm delivery, Curaçao, Caribbean

## Abstract

Sexually transmitted infections are one of the important risk factors for preterm delivery, which is among the important contributors to perinatal morbidity and mortality. The aim of this study was to assess the prevalence of *Chlamydia trachomatis* and *Neisseria gonorrhoeae* infections in women with imminent preterm delivery in Curaçao, an island of the Dutch Caribbean. All women from Curaçao with either preterm premature rupture of the membranes or preterm labor, common indications of imminent preterm delivery, and presenting at the Curaçao Medical Center between 15 November 2019 and 31 December 2020, were included in this single cohort study. Data were retrospectively collected from medical records. The presence of *Chlamydia trachomatis* and *Neisseria gonorrhoeae* was assessed by Cepheid GeneXpert ^®^ (Xpert) CT/NG assay (Sunnyvale, CA, USA). In the included cohort, the prevalence of *Chlamydia trachomatis* infection was 15.5% and of *Neisseria gonorrhoeae* infection was 2.1%. All patients infected with *Neisseria gonorrhoeae* were co-infected with *Chlamydia trachomatis*. The prevalence of *Chlamydia trachomatis* and *Neisseria gonorrhoeae* infections in patients with imminent preterm delivery in Curaçao is high. It is recommended to test all patients with imminent preterm delivery for these sexually transmitted infections and possibly consider testing all women in early pregnancy on the island.

## 1. Introduction

Preterm delivery, defined as delivery prior to 37 weeks of gestational age, is the highest contributor to perinatal morbidity and mortality worldwide, accounting for 70% of neonatal mortality [1,2,3]. Likewise, the risks of neonatal adverse outcomes, such as respiratory problems and necrotizing enterocolitis, decrease with the progression of gestational age [1,4]. Worldwide, rates of preterm delivery range from 5 to 18% [1,5]. About 40–45% of preterm deliveries are preceded by preterm labor and 25–30% by preterm premature rupture of membranes (PPROM) [6]. Preterm labor is defined as labor before 37 weeks of gestation, and PPROM is defined as the spontaneous rupture of the membranes before 37 weeks of gestation at least one hour before the onset of contractions. Most women with PPROM develop spontaneous contractions within several days, but in some, delivery can be delayed for weeks or months [6].

Vaginal infection, uterine overdistension (multiple gestations, polyhydramnios), short interval between pregnancies, previous preterm delivery, and cervical anomalies are some of the risk factors for developing preterm labor [1,2,4,5,6]. Genital tract infections, which are often asymptomatic, are present in 25–40% of preterm births and are seen as an important risk factor for developing both PPROM and preterm labor [6]. Vaginal infections can be caused by treatable sexually transmitted pathogens. *Chlamydia* (*C.*) *trachomatis* and *Neisseria* (*N.*) *gonorrhoeae* are two of the most common sexually transmittable infections (STIs). There is increasing evidence that *C. trachomatis* in particular can ascend from the lower to the upper genital tract and cause intrauterine infection [7]. Although study results vary, infection with *C. trachomatis* during pregnancy is associated with both PPROM and preterm labor, intrauterine fetal death, endometritis and conjunctivitis or pneumonia in newborns [3,8,9,10,11,12,13,14]. Infection with *N. gonorrhoeae* is significantly found to double the risk of preterm birth (OR 2) and is associated with chorioamnionitis, prematurity, and fetal growth retardation [15,16,17].

Studies have shown that the early detection and eradication of *C. trachomatis* during pregnancy could decrease the odds of preterm delivery [18]. So far, however, screening for STIs during pregnancy is not common practice in most countries, and testing and managing these infections is mostly performed in symptomatic patients. However, in certain populations, up to 80% of infections are asymptomatic [3,8]. This evidence suggests that the majority of infected pregnant women might carry pathogens while remaining undiagnosed or untreated. In Curaçao, a Caribbean Island within the Dutch Kingdom, the overall prevalence of infection with *C. trachomatis* in 1993 was reported to be 5% in the overall population and 10% in women under the age of 25 [19]. In another study carried on at a general practitioner practice between 1987 and 1991, the prevalence of *N. gonorrhoeae* was reported to be 0.4% in women between 15 and 64 years [20]. In the latter study, none of the infected women were pregnant at the time. Data on vaginal infections, including sexually transmitted ones, among the approximately 155,000 people living in Curaçao are scant and outdated [21]. To our knowledge, data among symptomatic women, such as women who experienced imminent preterm labor or spontaneous preterm premature rupture of membranes (SPPROM) in Curaçao, are also not available.

The aim of this study was to retrospectively investigate the prevalence of *C. trachomatis* and *N. gonorrhoeae* in patients with imminent premature delivery in Curaçao. Furthermore, a description of the patient characteristics and pregnancy outcomes of patients with imminent preterm delivery admitted at the Curaçao Medical Center between 15 November 2019 and 31 December 2020 was compiled.

## 2. Results

During the study period, 170 patients with imminent preterm delivery were admitted at the CMC and met the inclusion criteria. Table 1 summarizes the demographic and clinical characteristics of the patient population. Patients had a mean age of 28.4 years (SD 6.2) and a median BMI of 30 (IQR 25–35). Patients presented at a median gestational age of 34 weeks and 1 day; 77.1% of them presented with premature labor and 46.5% after spontaneous rupture of membranes (with or without contractions). Tocolysis and dexamethasone were started in 32.9% of patients. Obstetric history stated prior preterm delivery in 11.2% and early premature delivery in 5.9% of patients. Patients stayed a median of 3 days in the hospital.

Of the 170 patients with imminent preterm delivery, 148 (87%) were tested for *C. trachomatis* and 145 for *N. gonorrhoeae* (80%) infections (Table 2). Among them, 15.5% had a vaginal sample positive of *C. trachomatis* and 2.1% of *N. gonorrhoeae*, respectively. All of the three vaginal samples positive with *N. gonorrhoeae* were double infections with *C. trachomatis*. Moreover, in total, 77 vaginal swabs of 170 patients (41.8%) contained at least one tested microorganism. Table 2 shows the presence of infections reported during admission among 113 women. The prevalence of bacterial vaginosis was 10.7%, of *Trichomonas vaginalis* was 0.9%, of *Candida albicans* 17.8% and of *Streptococcus agalactiae* 26.4%. The most common coinfections were between *Streptococcus agalactiae* and *Candida albicans*, between bacterial vaginosis and *Candida albicans* and between *C. trachomatis* and *N. gonorrhoeae*. In 5% of 121 patients, a urinary tract infection was diagnosed. The data reported positive or negative values for urinary tract infection, and not always was a specific pathogen later cultured. Thus, the result of the cultured analysis was not included in this study.

Table 3 summarizes patients’ pregnancy outcomes. The median gestational age at delivery was 36 weeks and 1 day (33 + 3–37 + 0). In most patients (46.5%), delivery started with rupture of membranes, followed by spontaneous contractions in 35.9% of cases. In some cases (8.2%), the labor was induced, or a primary cesarean delivery (9.4%) was performed. Most patients delivered spontaneously (63.8%), in some cases, a vaginal assisted delivery was performed (1.7%), and 29.9% underwent a cesarean delivery. Neonates had a median Apgar score of 8 (7–9) after 1 min and 9 (8–10) 5 min after delivery, had a mean birth weight of 2426 g on the 46th percentile. Overall, 52.2% of neonates were of male gender. All 15 neonates born before 25 weeks of gestational age expired. Of the neonates born after 25 weeks of gestational age, nine (5.6%) expired.

## 3. Discussion

To our knowledge, this is the first study that reports a prevalence of genital infections in patients with imminent preterm delivery in Curaçao. We found a prevalence of 15.5% for *C. trachomatis* and 2.1% for *N. gonorrhoeae* infections.

Unfortunately, no data are available on the prevalence of *C. trachomatis* and *N. gonorrhoeae* in the general population in Curaçao during the same period. The previous study conducted in 1993 in Curaçao showed a lower point prevalence of *C. trachomatis* in the general population (overall, 5% in women and 10% in women under 25 years) [19]. The latest data regarding *C. trachomatis* prevalence have been published online by the Institute for Public Health Curaçao (Volksgezondheid Instituut Curaçao, VIC). Between 2008 and 2016, 9.0–14% of samples of women tested at ADC were positive for *C. trachomatis* [22]. The highest positivity rate (31.7%) was for the age group between 15 and 19 years [22]. The data are only from samples tested at the ADC and not at other laboratories in Curaçao, and it does not indicate where the samples were collected (hospital or general practitioner) or any clinical data (symptoms or proportion of women who were pregnant). However, in concordance with our data, it does show a high *C. trachomatis* positivity rate among women in Curaçao. A Dutch study conducted in the Netherlands showed also a similar high prevalence of *C. trachomatis* (16.2%) in pregnant (former) Antillean women living in the Netherlands [3]. These women were born on former Dutch Antilleans Islands in the Caribbean (Aruba, Bonaire, Curaçao, Saba, St. Eustacia and St. Maarten), with the majority coming from Curaçao. There are, however, several reasons that make comparing this group and our studied population difficult, such as cultural and geographic differences.

Different studies conducted in Latin America have shown a comparable prevalence of *C. trachomatis* infection of 11.2% (95% CI 7.3–17.1) but a lower prevalence of *N. gonorrhoeae* infection of 0.3% (95% CI 0.1–2.1) [23]. A direct comparison between studies describing STI prevalence in other Caribbean countries and Curaçao is also difficult due to major socioeconomic, cultural and behavioral differences [24]. Nevertheless, when comparing our data to the study conducted in Curaçao in 1993, it seems that, after 27 years, the prevalence of *C. trachomatis* infection has increased among certain groups of women residing in Curaçao [19]. For previous studies in Curaçao studying *N. gonorrhoeae* infection, the same conclusion can be drawn. The prevalence of *N. gonorrhoeae* infection in non-pregnant females between 1987 and 1991 was 0.4%, whereas in this study, it was 2.1% [20]. Differences in the laboratory detection methods used between the previous studies and this study (culture, enzyme-linked immunosorbent assay versus real-time PCR) might also explain this difference.

Vaginal infections data, including on STIs, are scarce from Curaçao, as they are for most parts of Latin America and the Caribbean [23,25]. Differences between socioeconomic, cultural, and behavioral factors, as well as practices of STI testing, call into question comparisons. Pregnant groups are an important source of information regarding STI status in low-risk groups that often do not get tested [25]. Therefore, more efforts to investigate the burden of, and manage, treatable pathogens associated with negative reproductive and pregnancy outcomes, including preterm delivery, are warranted [8,9,10]. Considering the data of this specific population and the high prevalence of *C. trachomatis* and *N. gonorrhoeae* in Curaçao previously reported, more interventions should be implemented to decrease these infection rates in high risk (pregnant women) and general population. Interventions such as screening during pregnancy, contact tracing, and sexual education in schools are warranted.

Because of the retrospective design of this study, it carries some limitations. The sample collection was not uniform, and only a small cohort of women were eligible to be investigated in this study. The rate of missing data was 15%; thus, not all women were tested. This was due to various reasons. For instance, if the women delivered soon after arrival, the physician might have forgotten to take the genital sample.

As mentioned before, studies have shown that screening for *C. trachomatis* and *N. gonorrhoeae* and subsequently treating their infection might improve pregnancy outcomes [10,26]. To date, the World Health Organization (WHO) recommends a syndromic approach with treatment only for symptomatic women [27]. Currently, at the gynecology department of CMC, it is in the local protocol to test for *C. trachomatis* and *N. gonorrhoeae* solely in symptomatic patients, for instance, women presenting with imminent preterm delivery. Therefore, no comparison with asymptomatic pregnant women could be retrospectively performed to determine any association between *C. trachomatis* infection and imminent preterm delivery in Curaçao. However, a multivariable sub-analysis showed that in this population, *C. trachomatis* or *N. gonorrhoeae* infection was not significantly associated with either birth weight or Fenton growth chart weight percentile (data not shown). Unfortunately, since most infections by *C. trachomatis* and *N. gonorrhoeae* are asymptomatic, the syndromic approach to detect and treat them might miss the majority of infections, also in pregnancy. Most importantly, prompt testing and the management of *C. trachomatis* and *N. gonorrhoeae* infections might decrease the odds of developing pelvic inflammatory disease, transmitting human immunodeficiency virus (HIV), developing infertility or other adverse effects, and might overall improve female reproductive health and prevent infections in the neonate such as neonatal conjunctivitis.

This study provides essential real-time clinical data on the sexual and reproductive health of Curaçao women collected at the only high-care obstetrics hospital on the island. Like the limited information available earlier, this study shows a high prevalence of *C. trachomatis* infection among women with imminent preterm delivery in Curaçao [19,20]. Routinely testing and treating *C. trachomatis* and *N. gonorrhoeae* infections in all pregnant patients in Curaçao might contribute to decreasing the rates of imminent premature delivery and neonatal conjunctivitis or lower the chance of developing long-term sequalae, not to mention the possible psychological burden. To further investigate this assumption, a prospective study according to the WHO standard protocol to assess the prevalence of gonorrhea and chlamydia among pregnant women in antenatal care clinics with larger patient population should be conducted [28].

## 4. Materials and Methods

### 4.1. Patients and Design

This single-center descriptive retrospective cohort study was performed at the Curaçao Medical Center (CMC) in Willemstad, Curaçao. The CMC opened on 15 November 2019 and is currently the only hospital on Curaçao treating pregnant women with medical indications. Patients were identified through admission records by the gynecology ward. All patients who were admitted between 15 November 2019 and 31 December 2020 with imminent preterm delivery with a gestational age between 16 weeks and 0 days and 36 weeks and 6 days were included. Imminent preterm delivery was defined as either 1. spontaneous preterm rupture of membranes or 2. preterm contractions with a cervical length <25 mm or cervical dilatation >1 cm. Non-resident patients who were transferred from another country to the CMC for medical help and patients who delivered elsewhere (first line maternity ward clinics) were excluded. This study was conducted after approval of the local institutional review board of the CMC (METC CMC) and after receiving a waiver of informed consent.

### 4.2. Data and Samples Collection and Testing

Demographic data, medical history, vital signs, microbiology results and data on pregnancy and delivery were extracted from the medical records of the CMC. The primary outcome was the presence of *C. trachomatis* and *N. gonorrhoeae* pathogens in samples collected with vaginal or cervical eSwab (flocked swab with 1 mL Amies transport medium, Copan Italia, Brescia, Italy). As per protocol of imminent preterm delivery, vaginal samples were collected at admission and were sent to the Analytical Diagnostic Center (ADC) medical laboratory. The presence of *C. trachomatis* and *N. gonorrhoeae* was assessed by Cepheid GeneXpert ^®^ (Xpert) CT/NG assay (Cepheid). Results were reported back as positive or negative in the patient electronic file. Secondary outcomes included: presence of other microorganisms via vaginal culture (for instance Candida albicans, Trichomonas vaginalis), real-time PCR positive for streptococcus agalactiae, presence of bacterial vaginosis (defined by Gardnerella vaginalis growth in culture), gestational age at delivery, onset of active labor, mode of delivery, birth weight, Apgar scores, and length of hospital stay. Per the local protocol of imminent preterm delivery, in patients under 34 weeks of gestational age, a transvaginal ultrasound was performed to measure cervical length rather than by digital examination of the cervix. Data on the presence or absence of urinary tract infection by urine stick and/or urine culture were also collected. Urinary tract infection was considered positive in the case of positive nitrite or leukocytes in urine stick or in the case of bacteriuria found in urine culture.

### 4.3. Statistical Analysis

Descriptive statistics were performed on the single cohort data, calculating mean values with standard deviation (SD), medians with interquartile ranges (IQR) and percentages. Statistical analyses were performed using IBM SPSS statistics 27 (SPSS Inc., Chicago, IL, USA).

## 5. Conclusions

The prevalence of *C. trachomatis* and *N. gonorrhoeae* infections in patients with imminent preterm delivery in Curaçao is high (15.5% and 2.1%, respectively). Considering the high prevalence in this population, we would recommend testing every patient with (imminent) preterm delivery in the island for these infections and other sexual transmitted pathogens such as *Treponema Pallidum* (syphilis). One could also consider routinely testing all patients in the first trimester of pregnancy, so the early treatment of these STIs could prevent adverse pregnancy outcome and adverse effects on female reproductive health, or routinely testing in the third trimester to reduce the risk of neonatal conjunctivitis. A prospective study is needed to investigate the effect of testing for and treating *C. trachomatis* and *N. gonorrhoeae* infections in the first trimester and make a cost–benefit analysis.

## Figures and Tables

**Table 1 pathogens-11-00670-t001:** Patient characteristics at admission among the 170 included patients.

Patient Characteristics	
Age [mean (SD)]	28.4 (6.2)
BMI [median (IQR)]	30.0 (25.0–35.0)
Gravidity [median (IQR)]	2 (1–4)
Parity [median (IQR)]	1 (0–2)
Medical indication at admission [*n* (%)]	
- Prior early premature delivery ^1^ - Prior premature delivery - Dichorionic diamniotic twin pregnancy - Monochorionic diamniotic twin pregnancy - Cervical insufficiency - Prior cesarean delivery - Prior hypertensive disorder in pregnancy - Other medical indication ^2^ - No medical indication	10 (5.9) 19 (11.2) 8 (4.7) 6 (3.5) 13 (7.6) 25 (14.7) 13 (7.6) 81 (47.6) 45 (26.5)
Gestational age at admission [median (IQR)]	34 + 1 (29 + 4–35 + 6)
Hospital length of stay, days [median (IQR)]	3 (2–6)
Vital signs at admission ^3^ [median (IQR)] - Tachycardia (*n* = 165) - Hypertension (*n* = 168) - Fever (*n* = 149)	53 (32.1) 18 (10.7) 5 (3.4)
Premature rupture of membranes [*n* (%)] - Hours with ruptured membranes [mean (SD)] Range	79 (46.5) 13.5 (31.2) 0–227
Dilation at admission (cm) [range] (*n* = 131)	2 (0–4)
Dilation >3 cm [*n* (%)]	64 (48.9)
Cervical length at admission (mm) [range] (*n* = 120)	0–45
Cervical length <25 mm [*n* (%)]	44 (36.7)
Contractions at admission [*n* (%)]	131 (77.1)
Tocolysis started [*n* (%)]	56 (32.9)

^1^ Spontaneous delivery between 16 and 24 weeks of gestational age. ^2^ Other medical indications included prior or current fetal growth restriction, congenital abnormalities, prior intrauterine fetal demise, advanced maternal age, maternal obesity, (gestational) diabetes, uterine myomatosis, hyperemesis gravidarum, asthma, blood loss during pregnancy, prior shoulder dystocia. ^3^ Tachycardia is defined as pulse > 100 bpm. Hypertension is defined as systolic blood pressure > 140 mmHg or diastolic blood pressure > 90 mmHg. Fever is defined as temperature > 38.0 degrees Celsius.

**Table 2 pathogens-11-00670-t002:** Prevalence of genital and urinary tract infections among tested patients.

Presence of Infection	*N* (%)
Any vaginal infection	77 (41.8)
*Chlamydia trachomatis* (*n* = 148)	23 (15.5)
*Neisseria gonorrhoeae* (*n* = 145)	3 (2.1)
Urinary tract infection (*n* = 121)	6 (5.0)
*Streptococcus agalactiae* (*n* = 144)	38 (26.4)
Other genital infections (*n* = 113)	
- *Candida albicans* - *Trichomonas vaginalis* - Bacterial vaginosis - *Staphylococcus hemolyticus* - *Proteus mirabilis* - *Streptococcus pseudoporcinus* - *Staphylococcus sacrophyticus* - *Staphylococcus aureus*	20 (17.8) 1 (0.9) 12 (10.7) 1 (0.9) 1 (0.9) 1 (0.9) 1 (0.9) 1 (0.9)

**Table 3 pathogens-11-00670-t003:** Outcome of pregnancy.

Pregnancy Outcome	*N* = 170 Pregnancies, *N* = 184 Neonates
Gestational age at delivery [median (IQR)]	36 + 1 (33 + 3–37 + 0)
Time between admission and delivery in days [median (IQR)]	2 (0–25)
Start of labor [*n* (%)]	
- Spontaneous contractions - Rupture of membranes - Induction of labor - Primary cesarean delivery	61 (35.9) 79 (46.5) 14 (8.2) 16 (9.4)
Mode of delivery [*n* (%)]	
- Spontaneous delivery - Vaginal assisted delivery (ventouse) - Cesarean delivery	124 (67.4) 4 (2.2) 56 (30.4)
Sex neonate [*n* (%)] (*n* = 183)	
- Male - Female	96 (52.2) 87 (47.3)
Birth weight in gram [mean (SD), range] - Percentile birth weight Fenton curve	2426 (868), 180–4150 46.0 (27.4)
Apgar scores [median (IQR)]	
- 1 min - 5 min	8 (7–9) 9 (8–10)
Neonate expired [*n* (%)] - <25 weeks of gestational age [*n* (%)] - >25 weeks of gestational age [*n* (%)] (*n* = 162)	15 (100) 9 (5.6)

## Data Availability

Data are available from the authors with the permission of the clinical center governing body/local institution and the Principal Investigator of the site.

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
