# Peer review of "Frequency of *Chlamydia trachomatis* and *Neisseria gonorrhoeae* in Patients with Imminent Preterm Delivery on the Island of Curaçao"

_pathogens, 2022, doi:10.3390/pathogens11060670_

Round 1

Reviewer 1 Report

The manuscript is an attempt to describe the epidemiological situation of STIs that has developed in Curaçao in relation to women with imminent preterm delivery. The high level of Chlamydia trachomatis infection was revealed in the patients cohort. The incidence of Neisseria gonorrhoeae are also turned to be high enough. The epidemiology of the STIs in general population of Curacao island referenced by the authors also indicated high values. The readers may wonder why is the situation arose and what are the activities for the future the authors can propose to decrease the STIs incidence not only in pregnant womens, but in general population? Besides, in conclusions section the authors propose 'Considering the high prevalence in this population, we would recommend to test every patient with (imminent) preterm delivery in the island for these infections.' however the WHO strategy consider also Treponema Pallidum testing, that is also one of the important reason for preterm delivery (https://www.paho.org/en/topics/sexually-transmitted-infections). I recommend the authors add to this sentence 'syphilis testing' besides the infections reported in the manuscript.

Author Response

> Reviewer 1:
> The manuscript is an attempt to describe the epidemiological situation of STIs that has developed in Curaçao in relation to women with imminent preterm delivery. The high level of Chlamydia trachomatis infection was revealed in the patients cohort. The incidence of Neisseria gonorrhoeae are also turned to be high enough. The epidemiology of the STIs in general population of Curacao island referenced by the authors also indicated high values.

>Thank you for taking the take to go over our manuscript and provide valuable feedback. 

> -The readers may wonder why is the situation arose and what are the activities for the future the authors can propose to decrease the STIs incidence not only in pregnant womens, but in general population?

>Thank you for your comment. In line 174-178, we wrote some advice for future implementations that would contribute to a decrease in the STI prevalence in the general population of Curacao. 

> -Besides, in conclusions section the authors propose 'Considering the high prevalence in this population, we would recommend to test every patient with (imminent) preterm delivery in the island for these infections.' however the WHO strategy consider also Treponema Pallidum testing, that is also one of the important reason for preterm delivery (https://www.paho.org/en/topics/sexually-transmitted-infections). I recommend the authors add to this sentence 'syphilis testing' besides the infections reported in the manuscript.

>Thank you for this valuable recommendation. Indeed syphilis is also an important reason for preterm delivery. We added this information in lines 260-261. 

Reviewer 2 Report

The authors conducted a retrospective study investigating the prevalence of a number of pathogens associated with vaginal infections in patients with imminent premature delivery. They found infections present in ~41% of patients. C. trachomatis was identified in ~15% of patients and N. gonorrhoae in ~2%.

The manuscript is well-written and the data is presented clearly. The results presented are not very surprising, as the authors note that vaginal infections are often found to be associated with imminent premature delivery.

A major downside to the study is that the authors do not know the background level of Ct or Gc infections in their population at the time the study was conducted. The data they cite for the Curacao background population (by their own admission) is rather dated. Their discussion concerning how Chlamydia and Neisseria infections are confirmed in the clinic has changed in the 30 years was on point, and very well may have influenced the change in rates of infection reported over that long time period.

Line 199-200: Rather than the prevalence in women in Curacao in general, the present study simply indicates that there is a high prevalence of infections in Curacao patients exhibiting symptomology (preterm labor). As this is to be expected in this particular patient population (25-40%, in the study cited by the authors), I do not believe that it can be used to infer that the background infection rates in the women in Curacao are ‘high’.

Author Response

> Reviewer2:

> The authors conducted a retrospective study investigating the prevalence of a number of pathogens associated with vaginal infections in patients with imminent premature delivery. They found infections present in ~41% of patients. C. trachomatis was identified in ~15% of patients and N. gonorrhoae in ~2%.

> The manuscript is well-written and the data is presented clearly. The results presented are not very surprising, as the authors note that vaginal infections are often found to be associated with imminent premature delivery.

Dear reviewer, thank you for the positive comments. Indeed the results are not very surprising however, in our opinion, it is very important to highlight those treatable infections are still highly prevalence in sequalae. 

> -A major downside to the study is that the authors do not know the background level of Ct or Gc infections in their population at the time the study was conducted. The data they cite for the Curacao background population (by their own admission) is rather dated. Their discussion concerning how Chlamydia and Neisseria infections are confirmed in the clinic has changed in the 30 years was on point, and very well may have influenced the change in rates of infection reported over that long time period.

>This is indeed one of the main reasons why we have started di work and feel it is important to report our findings. 

> -Line 199-200: Rather than the prevalence in women in Curacao in general, the present study simply indicates that there is a high prevalence of infections in Curacao patients exhibiting symptomology (preterm labor). As this is to be expected in this particular patient population (25-40%, in the study cited by the authors), I do not believe that it can be used to infer that the background infection rates in the women in Curacao are ‘high’.

Thank you for this feedback. We agree that it is important to clarify well that this data is from patients exhibiting symptomology (preterm labor), and not from general population. Our revision is written in line 205.